# Quality of evidence in the oral health international data: Contributions for a global profile

**Sophia Queiroz Marques dos Santos**[1☯]*, **Angelo Giuseppe Roncalli da Costa Oliveira**[2☯]

1 Master in the Postgraduate Program in Public Health, Dentistry Department, Federal University of Rio Grande do Norte, Natal, Rio Grande do Norte, Brazil, 2 Coordinator of the Postgraduate Program in Public Health, Dentistry Department, Federal University of Rio Grande do Norte, Natal, Rio Grande do Norte, Brazil

☯ These authors contributed equally to this work.
* soqueirozm@hotmail.com

**Data Availability Statement:** The data utilized is publicly available and can be accessed on the

## Abstract

### Introduction

The Oral Health Country/Area Profile Project (CAPP) is the largest global database on oral health, compiling information from 205 countries, including 193 members of the World Health Organization (WHO). Although this database is a source of information and provides an overview of global oral health, the extent to which it accurately reflects oral health in specific countries is uncertain.

### Objective

To analyze the quality of evidence underlying the global oral health profile provided by CAPP.

### Methods

The Appraisal tool for Cross-Sectional Studies (AXIS) was adapted and used to assess data extracted from the methods section of included records. The results were then analyzed using the Item Response Theory (IRT) to establish the weightings of each dimension. The score was assessed in relation to variables of interest: age group, year of the record, and geographic region.

### Results

The quality of oral health data showed polarization, with The quality of data from included documents varied according to age group analyzed, year of assessment, and geographic region. The Americas and Western Pacific regions demonstrated the highest quality of oral health data.

### Conclusion

The global oral health profile depicted by CAPP may not accurately reflect reality. The process for including data in the database needs to be reviewed to ensure its reliability.

website (https://mau.se/en/about-us/faculties-and-departments/faculty-of-odontology/oral-health-countryarea-profile-project–capp/), which is referenced throughout the text. The authors explicitly state that they downloaded a spreadsheet containing the data and document titles through this site. In an effort to clarify the level of evidence of the studies presented in the current research, these studies were identified as shown in the table in the supplementary material. The authors believe it is feasible to redirect to the files through the website and via the supplementary material, which contains access URLs. This ensures that the method is replicable and researchers can recover the data in a similar manner.

**Funding:** The author(s) received no specific funding for this work.

**Competing interests:** The authors declare that they have no conflicts of interest.

## Introduction

Oral health is a vital component of human well-being, influencing the quality of life and general health [1]. However, dental caries remains the most prevalent oral condition in the world, despite years of assessment, age, or geographic location [2]. The World Health Organization (WHO) considers oral health as a reflection of health problems and mitigating dental cavities is complex [2]. Furthermore, the availability of data about caries becomes crucial for the development of public policies that promote changes at the various levels that determine this condition [3].

Epidemiological surveys are a strategy to determine oral health status and the need for intervention among communities [4]. Understanding the methodological parameters used in those surveys is important to reducing information bias. The size and representativeness of the sample, sample selection method, the territory covered in the study, and the set of data collected are examples of factors contributing to the strength of a data source [5].

The Oral Health Country/Area Profile Project (CAPP) is a WHO Collaborating Center and held the largest database on global oral health, compiling information from 205 countries. The WHO Collaborating Center for Education, Training, and Research at the Faculty of Odontology of Malmö, Sweden, developed the CAPP in 1995, and since then, has offered information on oral health conditions worldwide. For its development, consultations were held with WHO Collaborating Centers, such as the Non- Communicable Diseases Group (Geneva, Switzerland), and other WHO collaborators worldwide. The CAPP aims to present information on dental diseases and oral health services for different countries and areas [6].

The CAPP database is composed of various records with variables related to the year when the study was developed, age group and country [6, 7]. However, the inclusion criteria for the records to be part of the database are still unclear. Although epidemiological surveys on oral health are the gold standard for gathering data, they are time-consuming and high cost [4]. In this sense, using information previously collected facilitates database maintenance but raises a debate on data quality and the need for international inclusion criteria. Thus, the validity and applicability of the results obtained through the analysis of the data available in CAPP are influenced by the quality of evidence.

This study aimed to evaluate the quality of evidence in the documents available in the CAPP database, seeking to understand their methodological strength and data reliability. Our objective was to provide a critical and in-depth overview of the evidence underpinning the global oral health profile, addressing the gap in assessing the reliability of one of the largest databases in the world. Additionally, we aimed to identify gaps, potential biases, and ways to optimize the use of the database as a reference tool for global oral health, with the goal of improving the direction of global oral health strategies.

## Materials and methods

### Study design and data extraction

This analytical study used secondary data from the CAPP database. We downloaded data from the official CAPP website (https://capp.mau.se/download) in September of 2023 and exported it to Microsoft Excel®.

The CAPP database contains oral health information referred to specific years and countries, including the index of decayed, missing, and filled teeth (DMF-T for permanent teeth and dmf-t for deciduous teeth) and the percentage of cavity-free teeth. The references endorsing the CAPP database mainly comprise published studies or research reports. Those were the main materials used to analyze the quality of evidence available in the database.

A total of 1,222 records about caries data were retrieved in the CAPP database, all divided into different age groups and country. Each record was derived from a study or report and might have been related to other countries and age groups or not, which means that the number of records was probably lower than the number of documents.

The CAPP database contains the variables mentioned before (country, year of assessment, age group), but socio-economic indicators were also searched when available to establish a profile of countries and the factors influencing the respective quality of evidence.

The quality of the records also may vary according to the different WHO regions. These regions were defined to decentralize the geopolitical nature of administrative centers and facilitate the coordination of WHO activities at a global level: (1) Americas (PAHO/PAHO); (2) Southeast Asia (SEARO); (3) Europe and Central Asia (EURO); (4) North Africa and the Middle East (EMRO); (5) the Western Pacific (WPRO), and (6)Africa (AFRO)7.

## Data assessment tool

Data was screened in full text by one researcher, who also rated it according to the quality of evidence. We adapted the Appraisal tool for Cross-Sectional Studies (AXIS) developed by Downes et al. (2016) [8]. The tool distinguished itself from other quality assessment instruments due to its up-to-date nature and its broad applicability. This justifies its selection by the authors. The records were identified by title, country, and year. Each record was assessed using the AXIS dimensions, and the final score was determined by the sum of each dimension. The original version of the AXIS has 20 dimensions, but only nine (those related to methodology) were considered in the adapted version (Table 1). Dimensions were classified as a dichotomous variable (yes or no).

This model of unweighted scores disregards that some parameters are harder to meet or influence the quality of the evidence more than others. We used interdependence analysis to adjust score values because it is a valuable discriminatory technique. Data was submitted to the Item Response Theory (IRT) analysis to measure the impact of each AXIS dimension and provide a proper estimation of the weight's dimension. For each record, the final score of the AXIS was calculated from the weighted average using the following equation (D = dimension and W = weight):

$$AXIS\ score = \frac{(D_1 \times W_1) + (D_2 \times W_2) + (D_n \times W_n)}{(W_1 + W_2 + W_n)}$$

**Table 1. Characterization of the AXIS according to dimensions and scores.**

| Dimension | Score |
|---|---|
| D1- Is the aim of the study clear? | 1.0 |
| D2- Is the study design appropriate? | 1.0 |
| D3- Were the risk factors and outcome variables appropriately measured considering thestudy aim? | 1.0 |
| D4- Was the sample size justified? | 10 |
| D5- Was the target population clearly defined? (Is it clear to whom the study referred to?) | 1.0 |
| D6- Were measures taken to address and categorize non-responders? | 1.0 |
| D7- Was the sample based on an appropriate representation of the target population under investigation? | 1.0 |
| D8- Was the recruitment process representative of the target population being investigated? | 1.0 |
| D9- Was the study method used previously validated? | 1.0 |
| **Total** | **9.0** |

## Statistical analysis

The AXIS dimensions were analyzed considering the absolute and relative frequencies considering WHO region, year of the study, and age group as variables. For the calculation of the final AXIS score, discriminatory analysis of each dimension was performed using the Item Response Theory (IRT). Lastly, the final AXIS score was correlated with the variables of interest and mapped according to global distribution. Data were plotted in a georeferenced database, and map was generated using QGIS software (version 3.34).

## Results

From the 1,222 records retrieved, 1,004 were available for full text reading. After excluding duplicates, 890 records were included in the analysis (Fig 1). Of these, more than half (55%) were from the Americas or European continent, 44.72% were added to the database between 2015 and 2009, and 59.74% predominantly assessed children aged from zero to 12 years.

The frequencies of each AXIS dimension considering geographic location are shown in Table 2. Results were described considering two perspectives: vertically, each dimension can be analyzed alone and compared to other dimensions in relation to the effectiveness within the same category; horizontally, a comparison of the performance of different categories is related to the effectiveness of each dimension.

Considering the WHO regions, Africa and Southeast Asia showed the worst performances, particularly in the dimensions related to the adequacy of the population base and previous validation of the method used for analysis (Table 2). Regarding the year, a higher score was observed for recent records (from 2016 to 2022, Fig 2). Lastly, the age group from 0 to 12 years also showed better scores (Fig 2).

The frequency tables demonstrate disparities between the AXIS dimensions. Based on the reading of the record, some dimensions are more relevant to the final score. The use of IRT allowed individual identification of the discriminatory power of each dimension. This data can be used as weight and readjust the final score.

The dimensions with the highest discriminatory coefficient values were (a) whether the study design was adequate (4.19; CI 95% 1.78 to 6.60), (b) whether the sample base derived from a target population (3.42; CI 95% 2.61 to 4.22), and (c) whether the sampling method allowed the selection of representative individuals of the target population (5.27; CI 95% 2.98 to 7.56) (Table 3).

The final AXIS score expressed the overall quality of the evidence: the higher the score, the better the evidence. The AXIS score values were then analyzed considering the different characteristics of the countries (Fig 2).

## Discussion

The present study analyzed the quality of information available in the CAPP database using an adapted version of the AXIS related to some variables. Regarding age groups, the best scores were obtained by records including children and young adults, especially those aged between 15 and 19 years. This finding may be related to the recommendation of assessing the behavior of the permanent dentition a few years after exposure to the oral environment. In addition, the WHO advocates mapping oral health status until 12 years old [9]. The suggestion of analyzing the performance of the permanent dentition a few years after its exposure may have motivated the inclusion of groups ranging from 15 to 19 years old [9].

The categorization of our results into three time periods showed that data quality improved over time. Few countries were repeated in the three time periods, indicating that the improvement is limited to specific points. Between 2003 and 2009, American countries, especially

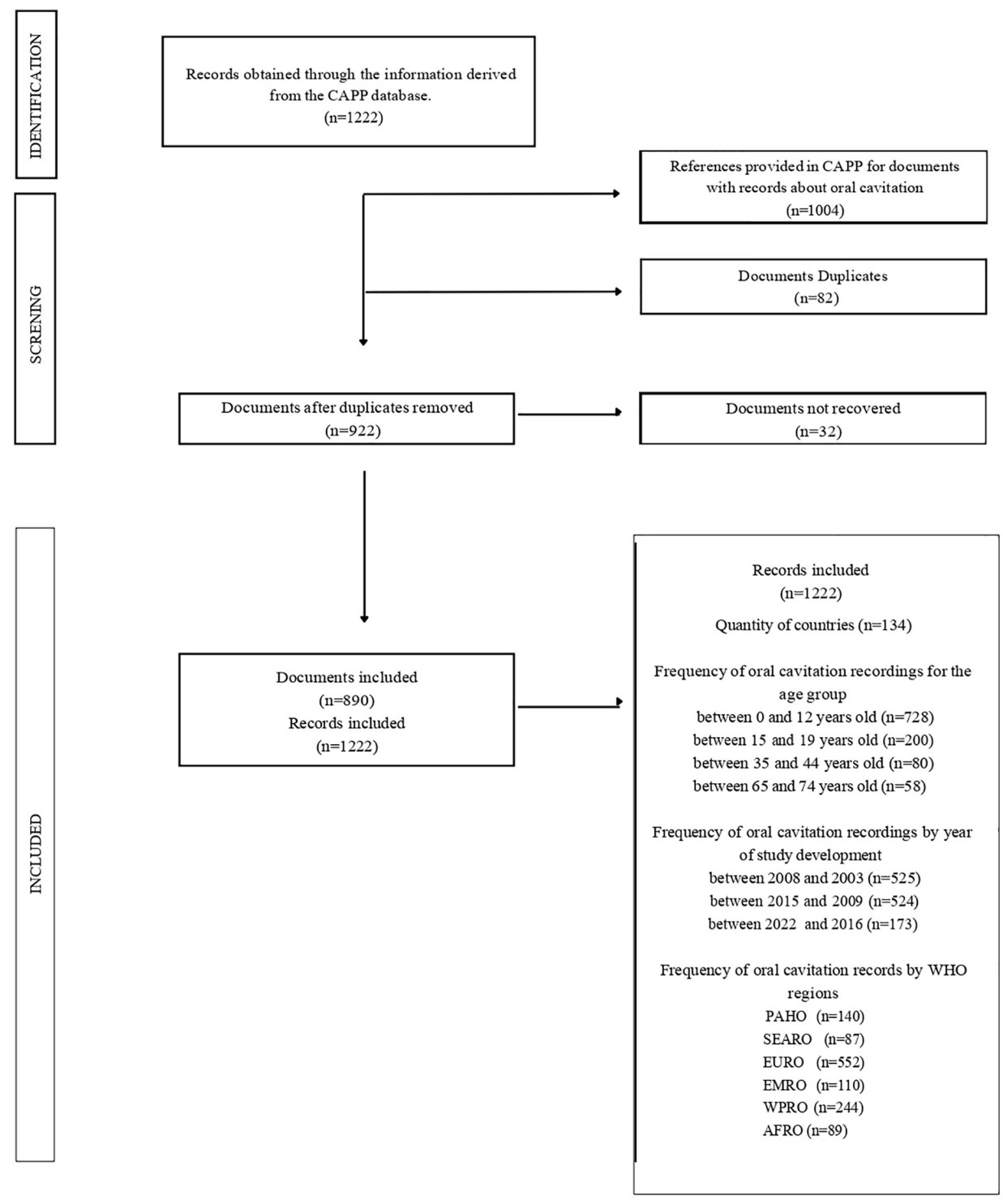

**Fig 1. Flowchart of the quantitative analysis of documents based on all records collected for different age groups, available in the CAPP database.**

**Table 2. Distribution of AXIS dimensions according to the World Health Organization (WHO) region.**

| | WHO area | | | | | | | | | | | | |
|---|---|---|---|---|---|---|---|---|---|---|---|---|---|
| | Africa | | Americas | | Eastern Mediterranean | | Europe | | Southeast Asia | | Western Pacific | | Total | |
| AXIS dimensions | n | % | n | % | n | % | n | % | n | % | n | % | n | % |
| D1- Is the aim of the study clear? | 85 | 95.5 | 116 | 82.9 | 106 | 96.4 | 452 | 81.9 | 40 | 46.0 | 171 | 70.1 | 970 | 79.4 |
| D2- Is the study design appropriate? | 85 | 95.5 | 107 | 76.4 | 106 | 96.4 | 446 | 80.8 | 40 | 46.0 | 171 | 70.1 | 955 | 78.2 |
| D3- Were the risk factors and outcome variables appropriately measured considering the study aim? | 86 | 96.6 | 118 | 84.3 | 108 | 98.2 | 499 | 90.4 | 40 | 46.0 | 200 | 82.0 | 1051 | 86.0 |
| D4- Was the sample size justified? | 42 | 47.2 | 77 | 55.0 | 24 | 21.8 | 242 | 43.8 | 16 | 18.4 | 112 | 45.9 | 513 | 42.0 |
| D5- Was the target population clearly defined? (Is it clear to whom the study referred to?) | 83 | 93.3 | 101 | 72.1 | 94 | 85.5 | 417 | 75.5 | 32 | 36.8 | 159 | 65.2 | 886 | 72.5 |
| D6- Were measures taken to address and categorize non- responders? | 1 | 1.1 | 1 | 0.7 | 0 | 0.0 | 17 | 3.1 | 0 | 0.0 | 4 | 1.6 | 23 | 1.9 |
| D7- Was the sample basedon an appropriate representation of the targetpopulation under investigation? | 19 | 21.3 | 60 | 42.9 | 43 | 39.1 | 186 | 33.7 | 7 | 8.0 | 57 | 23.4 | 372 | 30.4 |
| D8- Was the recruitment process representative of the target population beinginvestigated? | 39 | 43.8 | 80 | 57.1 | 66 | 60.0 | 266 | 48.2 | 7 | 8.0 | 102 | 41.8 | 560 | 45.8 |
| D9- Was the study method used previously validated? | 12 | 13.5 | 49 | 35.0 | 38 | 34.5 | 196 | 35.5 | 3 | 3.4 | 104 | 42.6 | 402 | 32.9 |
| TOTAL | 89 | 100.00 | 40 | 100.0 | 110 | 100.0 | 552 | 100.0 | 87 | 100.0 | 244 | 100.0 | 1222 | 100.0 |

Latin America, improved their data quality. The Asian continent also showed significant improvement, even though restricted to some regions, especially in South Asia. From 2009 to 2016, a trend towards maintaining data quality was demonstrated in North America, mainly in Canada, and improvements were observed in North Asia and centered on Russia. Africa and Europe showed subtle changes, and scores barely changed over the three time periods.

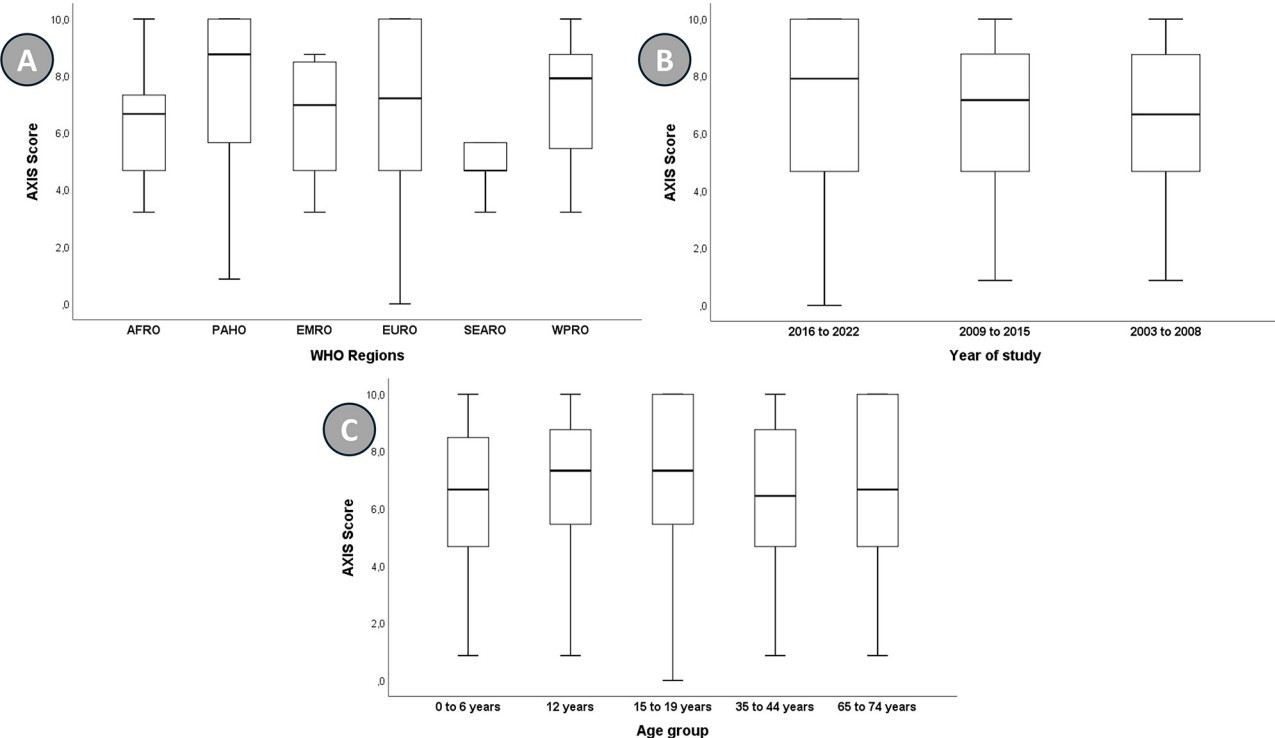

**Fig 2. Box-plot of the scores of the Appraisal tool for Cross-Sectional Studies (AXIS).** (A) World Health Organization (WHO) regions; (B) year of the study; (C) age group.

**Table 3. Discrimination coefficient for each item using the Item Response Theory (IRT) analysis.**

| Dimension | Discriminatory coefficient (CI 95%) | p-value |
|---|---|---|
| D1- Is the aim of the study clear? | No variation | - |
| D2- Is the study design appropriate? | 4.19 (1.78 to 6.60) | 0.001 |
| D3- Were the risk factors and outcome variables appropriately measured considering the study aim? | No variation | - |
| D4- Was the sample size justified? | 2.16 (1.77 to 2.54) | < 0.001 |
| D5- Was the target population clearly defined? (is it clear to whom the study referred to?) | 2.38 (1.72 to 3.04) | < 0.001 |
| D6- Were measures taken to address and categorize non-responders? | 1.59 (0.97 to 2.21) | < 0.001 |
| D7- Was the sample based on an appropriate representation of the target population under investigation? | 3.42 (2.61 to 4.22) | < 0.001 |
| D8- Was the recruitment process representative of the target population being investigated? | 5.27 (2.98 to 7.56) | < 0.001 |
| D9- Was the study method used previously validated? | 1.66 (1.36 to 1.95) | < 0.001 |

The quality of the records also varied according to the six distinct WHO regions. These regions were not delineated based on socioeconomic stratification, but rather to group countries with similar characteristics [6].

The Americas and Western Pacific regions encompass the major global power and had the highest mean scores of the adapted AXIS. The United States and Canada have seasonal oral health data since the mid-1990s [10, 11]. In Latin America, this scenario was only established around the 2000s, initially in Brazil, Cuba, and Mexico [12–14]. East Asia is a part of the Western Pacific region and had prominent national oral health surveys since the 1970s (Australia) and 1980s (China and Japan) [15, 16]. The data related to all countries mentioned in this paragraph were similar to national epidemiologic surveys and achieved higher scores.

The worst scores were observed in Africa and Southeast Asia. Most countries in Africa and India, Vietnam, Brunei, and Myanmar (South Asia) showed similar scenarios. First, only some studies were retrieved from these regions, and when available, had limited geography coverage [17–19], superficial content [17–19], and failed to show continuity [17–19], factors that compromised their quality and hindered use as a health planning tool.

Research funding for these countries exists based on the WHO initiatives in Africa that began in the 1980s but did not grow over time [20]. Therefore, resources were not able to meet the needs of monitoring the oral health situation or expand their territory representativeness [21].

The European scores were similar to the favorable medians, which is justified by historical epidemiological surveys. For instance, the United Kingdom and Germany have data from the 1930s and 1980s, respectively [22, 23]. Other countries of this group belong to Central Asia and were included in the former Soviet Union, showing scarce data retrieved [24]. On the other hand, North Africa and the Middle East had average results that align with low data quality, probably due to the marked geopolitical conflicts and extreme concentration of wealth [25]. The retrieval of data on oral health was limited in these contexts [26].

The Asian continent is known for its contrasts and was divided into four regions. The South of Asia belongs to the SEARO, is highly populated, with precarious living and health conditions, and has little development of health structures and public information [27]. The Middle East belongs to the EMRO and was marked by intense conflicts [28]. Central Asia, countries from the former Soviet Union, and part of Russia belong to the EURO, a conflict-ridden territory, difficult to interfere with, and with low accessibility to public data [29]. Last, East Asia belongs to the WPRO and shows extensive economic and scientific development with a better public health structure [30].

Countries that stand out in their respective regions, such as the United States and Canada in North America, Australia and Japan in East Asia, and Germany and the United Kingdom in Europe, achieved the best results due to high GDPs, good Human Development Index (HDI) scores, and well-developed health systems, particularly universal ones for Germany, the United Kingdom, Japan, and Australia. These countries have historically invested in robust health infrastructure and access to oral health services, which contributes to organizational interest—from a management perspective—in extensive and high-quality data collection. For example, Canada has had seasonal oral health data since the mid-1990s, reflecting a solid data foundation and continuous monitoring. In contrast, many African countries, such as those in Sub-Saharan Africa, face significant challenges in terms of infrastructure and health funding. The low GDP and HDI of these countries limit their ability to develop and maintain effective oral health systems, resulting in limited and poor-quality data. Additionally, for the study's assessment context, the lack of continuity and geographical coverage in studies from these regions further compromised the quality of available data. The lack of growth in research funding, despite WHO initiatives since the 1980s, also hinders the expansion of territorial representativeness and data collection improvement, reflecting a significant gap compared to regional powerhouse.

This study also provided a detailed analysis of the AXIS tool, mainly based on the interdependence analysis using the IRT model. Three crucial dimensions of AXIS were revealed: (1) the adequacy of the study design, linked to internal validity, reliability, and results representativeness [29]; (2) the sample drawn from an adequate population base and size, which was linked to the ability to bring findings closer to reality [30, 31], and (3) the population representativeness, ensuring the illustration of various social groups [32].

The study design is an important methodological aspect [33]. Aspects related to participant selection, research question, and theoretical framework were fundamental during the development of this study [34]. Furthermore, an appropriate design ensures appropriate scientific communication, transparency, and reproducibility of the study, indicating that the record strengths outweigh the limitations to achieving objectives [35].

The sample size is fundamental to guarantee validity since inadequate sample sizes lead to biases and erroneous conclusions. Moreover, an adequate sample size contributes to study reliability, lessening the influences of random variations and allowing greater applicability [36]. Researchers should pay attention to the planning and justification of the sample size to ensure their results are compatible with the studied scenario. Larger samples increase the probability of statistically significant differences and are more generalizable [37].

The methods used for sample selection should ensure a representative sample of the population under investigation and were more important than the sample size in this study. This is justified by the possibility of generalizing the results, but also reducing bias, external validation and reducing sampling error [38]. A correct sample selection is essential for precision, helping to obtain adequate results and efficient decisions in public health. Using the AXIS tool indicated that large-scale studies tend to be more robust to meet the tool dimensions, while smaller studies related to restricted territories may not prioritize these principles. However, large-scale research is expensive and depends on political factors, which explains the different availability of documents across territories [39] and justifies the diversity in terms of quality of the evidence.

The points discussed above corroborate the literature. The study of Kassebaum [1] showed the global prevalence and incidence of decay over the last 20 years. However, there is a possibility that their conclusions may be inconsistent, considering that the quality and representativeness of the data were not criteria for calculating global estimates. In this sense, the findings may have under or overestimated the real situation.

Specific scenarios illustrate the consequences of mistaken estimates. For example, although Brazil has seasonal national epidemiological surveys [40], the last survey available on the database reported some data from the state (federative unit) level [41]. Similar occurred for China, which last national oral health survey happened in 2017 [42], but its DMF-T index was derived from a provincial study from the same year [43]. Last, data about Italy dates back to 2016 and was restricted to children in Campania, the southern region [44]. However, the second national epidemiological survey was performed in 2017 for children aged up to 12 years [45], providing high-impact data that was not included or available in the CAPP database.

To address the issue of low-quality data production, it is crucial not only to invest in oral health research, with investments guided by epidemiological criteria, but also to ensure that the database adopts stringent criteria for selecting studies. These criteria should focus on the applied methodology and the representativeness of the studies. Even if a study is well-conducted, it may not accurately reflect the target population, which compromises the generalization of results and the production of research based on these data.

As strengths, the authors highlight the importance of considering the quality of evidence when including studies in the CAPP database. This consideration not only supports the planning of public health strategies, which should be developed based on accurate oral health data, but also the production of reliable evidence. It underscores the need for literature reviews and the development of new studies concerning the oral health landscape, with attention to the properties of the data used.

Limitations permeate the research development scenario, which is extremely complex in both operational and methodological aspects. This justifies the development of various studies with simpler methodological approaches. Furthermore, it highlights the importance of studies that examine social contexts, emphasizing that the criterion of sample representativeness can be as valuable as the final object of analysis. The uneven quality of data in the CAPP database and the use of secondary sources compromise potential conclusions. The documents used for the analysis had a certain geographical concentration, which limits the scope of temporal comparison in the writing of the work. However, its innovative and necessary nature justifies the continuation of the research.

## Conclusion

This study assessed the quality of evidence of records on the CAPP database, a global database of oral health. We sought to understand the methodological strength of the records and their data reliability. Significant variations were observed in data quality, with a positive impact on the age group and improvements being noticed over time. Challenges were evidenced in data from specific geographic regions. Criteria for including records in the database and periodic revision of the records in the CAPP are essential to ensure data accuracy and effectively guide actions that promote global oral health.

## Supporting information

**S1 File.**
(DOCX)

## Author Contributions

**Data curation:** Sophia Queiroz Marques dos Santos.

**Formal analysis:** Sophia Queiroz Marques dos Santos.

**Investigation:** Sophia Queiroz Marques dos Santos.

**Methodology:** Angelo Giuseppe Roncalli da Costa Oliveira.

**Software:** Sophia Queiroz Marques dos Santos.

**Supervision:** Angelo Giuseppe Roncalli da Costa Oliveira.

**Writing – original draft:** Sophia Queiroz Marques dos Santos.

**Writing – review & editing:** Angelo Giuseppe Roncalli da Costa Oliveira.

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
