## [Decision Letter · Decision Letter 0]

16 Sep 2024

PONE-D-24-29739Quality of evidence in the oral health international data: contributions for a global profilePLOS ONE

Dear Dr. Queiroz Marques dos Santos,

Thank you for submitting your manuscript to PLOS ONE. After careful consideration, we feel that it has merit but does not fully meet PLOS ONE’s publication criteria as it currently stands. Therefore, we invite you to submit a revised version of the manuscript that addresses the points raised during the review process.

We look forward to receiving your revised manuscript.

Kind regards,

Vanessa Carels

Staff Editor

PLOS ONE

2. Please note that your Data Availability Statement is currently missing the repository name and/or the DOI/accession number of each dataset OR a direct link to access each database. If your manuscript is accepted for publication, you will be asked to provide these details on a very short timeline. We therefore suggest that you provide this information now, though we will not hold up the peer review process if you are unable.

3. We note that Figure 3 in your submission contain [map/satellite] images which may be copyrighted. All PLOS content is published under the Creative Commons Attribution License (CC BY 4.0), which means that the manuscript, images, and Supporting Information files will be freely available online, and any third party is permitted to access, download, copy, distribute, and use these materials in any way, even commercially, with proper attribution. For these reasons, we cannot publish previously copyrighted maps or satellite images created using proprietary data, such as Google software (Google Maps, Street View, and Earth). For more information, see our copyright guidelines: http://journals.plos.org/plosone/s/licenses-and-copyright.

a. You may seek permission from the original copyright holder of Figure 3 to publish the content specifically under the CC BY 4.0 license. 

Reviewers' comments:

Reviewer's Responses to Questions

**Comments to the Author**

1. Is the manuscript technically sound, and do the data support the conclusions?

Reviewer #1: Yes

Reviewer #2: Partly

2. Has the statistical analysis been performed appropriately and rigorously? 

Reviewer #1: Yes

Reviewer #2: Yes

3. Have the authors made all data underlying the findings in their manuscript fully available?

Reviewer #1: Yes

Reviewer #2: Yes

4. Is the manuscript presented in an intelligible fashion and written in standard English?

Reviewer #1: Yes

Reviewer #2: Yes

5. Review Comments to the Author

Reviewer #1: Thank you for the opportunity to review this work. I have a few suggested revisions to strengthen it.

Abstract

Page 1, line 28-29: sentence seems to end abruptly, please revise

Methods

Page 4, lines 96 to end of page: can you please add some clarification on why AXIS was specifically chosen, if other frameworks were considered, what are the strengths of the framework and its limitations. Some of these details can be included later on in your discussion but good to have that information available.

Results

Page 7: there's a leftover comment on your PDF submission, please carefully check your manuscript for any edits/tracked changes etc before resubmitting.

Discussion

General comments

a. Can you please provide some explanation on the observed variations between countries/regions? At the moment, your discussion simply repeats key findings, there's no critical reflection on why some regions might be performing better/worse. This is a missed opportunity.

b. There are no clear strengths/limitations (especially the latter) when it comes to this work, can you please reflect (critically) on your work and share strengths/limitations.

Reviewer #2: The study addresses an important issue by evaluating the quality of data in the CAPP database, which is crucial for global oral health assessments. By critically assessing the quality of evidence, it fills a gap in ensuring the reliability of one of the largest oral health datasets globally. The objective of the study—to assess the quality of evidence in the CAPP database—is both clear and important. The CAPP database is the largest global oral health database, and understanding the reliability of its data is crucial for international health policymaking. The study’s relevance is underscored by its potential to improve global oral health strategies by identifying gaps in data reliability.

The use of a standardized tool, the Appraisal tool for Cross-Sectional Studies (AXIS), provides a systematic and validated framework for evaluating the data. The use of Item Response Theory (IRT) to weigh different dimensions of the data's quality adds an extra layer of analytical depth, ensuring that the most important aspects (e.g., study design, population representativeness) are given more weight.

However, the paper could benefit from more explicit aims regarding specific interventions for database improvement.

1. The rationale to conduct this analysis needs to be highlighted in the introduction.

2. Sampling and Data Source: The study relies exclusively on secondary data from the CAPP database, which inherently limits control over data quality. The authors acknowledge this but do not provide sufficient insight into how data quality issues might skew specific findings or how to mitigate this in future studies.

3. Geographical Representation: The paper's critical weakness lies in the uneven geographical representation of data quality. While this is a limitation of the CAPP database, the study could have provided more solutions for rectifying this issue. The lack of continuous and up-to-date data in certain regions, particularly in Africa and Southeast Asia, further weakens the generalizability of the findings.

4. Interpretation of Results

- Quality of Data by Region: The results point to significant variation in data quality across different regions, with the Americas and Western Pacific regions having better scores than Africa and Southeast Asia. This analysis is useful, but the paper falls short in discussing the potential socio-political factors behind these disparities. More in-depth exploration of why certain regions perform better or worse could add value to the analysis.

- Impact of Age Groups and Time Periods: The study finds that records focusing on children and young adults and more recent studies have higher quality data. However, the reasons for this are not fully explored. The authors suggest that this may be due to WHO guidelines focusing on younger age groups but stop short of a deeper analysis.

5. Study Limitations : The authors acknowledge key limitations such as the uneven quality of the CAPP data and the use of secondary sources. However, there are additional limitations that could have been addressed more thoroughly:

- Lack of In-Depth Analysis on Regional Disparities: The paper identifies disparities in data quality but does not propose robust solutions for addressing these gaps. For example, the study could have suggested methodologies for improving data collection in underrepresented regions.

- No Validation of Data from the Original Sources: The study does not attempt to cross-validate the data included in the CAPP database with other data sources, which could have strengthened the findings. Relying solely on the CAPP database limits the study’s capacity to assess the true reliability of the data.

6. Practical Implications and Recommendations: While the study provides a valuable critique of the CAPP database, it lacks actionable recommendations for policy-makers. Suggestions such as implementing stricter inclusion criteria, improving data collection in underrepresented regions, or fostering international collaboration to enhance the database would have been beneficial.

6. PLOS authors have the option to publish the peer review history of their article (what does this mean?). If published, this will include your full peer review and any attached files.

Reviewer #1: No

Reviewer #2: No

---

## [Author Response · Author response to Decision Letter 0]

23 Sep 2024

We appreciate all the feedback provided. We acknowledge that the reviewers raised extremely relevant points, and we will address each one individually. We are committed to refining the material to contribute effectively and accurately to the literature on the validation of oral health information.

Reviewer 1:

Comment 1: Page 1, lines 28-29: The sentence appears to end abruptly. Please review. 

Response: The authors thank the reviewer for this correction. We agree and have made the necessary change to improve comprehension. (Modified in the text: Page 1, lines 28-29)

Comment 2: Page 4, lines 96 to the end of the page: Could you please add some clarification on why AXIS was specifically chosen, if other frameworks were considered, and what are the strengths and limitations of the framework? Some of these details could be included later in your discussion, but it is good to have this information available. 

Response: The authors found it appropriate to include the justification for using the tool in the Methods section. (Modified in the text: Page 4, lines 100-102). We did not find it necessary to include a discussion of other evaluation tools as the AXIS tool was notably distinguished for being current and broadly applicable.

Comment 3: Page 7: There is a remaining comment in your PDF submission. Please check your manuscript carefully for any tracked changes or other issues before resubmitting.

Response: The authors apologize for the inconvenience. The empty comment has been removed, and the corresponding location now has a notation for verification. (Modified in Table 2, after line 165, page 4)

Comment 4: a. Could you please provide some explanation for the variations observed between countries/regions? Currently, your discussion merely repeats the main findings without any critical reflection on why some regions may perform better or worse. This is a missed opportunity. b. There are no clear strengths/limitations (especially the latter) concerning this work. Could you reflect (critically) on your work and share its strengths/limitations?

Response: a. The authors consider this feedback extremely pertinent. We discussed factors that might locally affect the variation in continental and regional scores but lacked a conclusion on the determinants of data production conditions. Consequently, a paragraph has been added to address this. (Modified in the text, lines 250-267, Pages 10-11) b. The authors acknowledge that the final paragraphs were reduced in clarity during translation. Key points required revision due to their importance for concluding the argument and impact on the science, so they have been rewritten. (Modified in the text, lines 317-322, 323-331, Pages 12-13)

Reviewer 2:

Comment 1: The rationale for conducting this analysis needs to be highlighted in the introduction. 

Response: The authors recognize a translation error that diminished the previously elaborated rationale and sincerely apologize. The paragraph has been rewritten to clearly state the justification for the study based on the literature gap and social impact. (Modified in the text, lines 65-71, Page 3)

Comment 2: The study relies exclusively on secondary data from the CAPP database, which inherently limits control over data quality. The authors acknowledge this but do not provide sufficient insights into how data quality issues may distort specific findings or how to mitigate this in future studies. 

Response: The authors are committed to studying the database in detail. The quality of evidence is part of a series of articles being developed, with this study serving as the starting point. Data quality indeed distorts oral health records, and a subsequent article will focus exclusively on analyzing documents considered well-rated by the AXIS tool, offering a comparative overview between the results obtained and previous studies. We hope to clarify why this topic was not addressed in the current study. Additionally, the need to counteract the production of low-quality data is emphasized not only through investment in oral health research but also through adopting selection criteria based on methodology and study representativeness. Sometimes a well-developed study does not represent the target population, affecting the generalization of results. (Modified in the text, lines 310-316, Page 12)

Comment 3: Geographic representation: A critical weakness of the article is the uneven geographic representation of data quality. Although this is a limitation of the CAPP database, the study could have provided more solutions to address this issue. The lack of continuous and updated data in certain regions, particularly in Africa and Southeast Asia, further weakens the generalization of findings. 

Response: The authors agree with this observation and highlight the geographical concentration as a limiting factor in the final paragraphs. We attempted to include as much data as possible, analyzing studies from 24 years ago to the most recent ones. While temporal comparison is not observed in all scenarios, what has been reported is significant. Some solutions were considered for the absence of data, but to ensure reliability, we worked only with the original data collected. Regional grouping was one attempt to observe if regions evolved differently over the years. Future studies have partially addressed this by weighting the article scores based on population numbers, aiming to clarify territorial representativeness, though it was not generalized across all territories within a region.

Comment 4: a. Data Quality by Region: The results indicate significant variation in data quality across regions, with the Americas and Western Pacific regions having better scores compared to Africa and Southeast Asia. This analysis is useful, but the article is insufficient in discussing the potential sociopolitical factors behind these disparities. A more in-depth exploration of why certain regions perform better or worse could add value to the analysis. b. Impact of Age Groups and Time Periods: The study concludes that records focusing on children and young adults and more recent studies have higher quality data. However, the reasons for this are not fully explored. The authors suggest that this may be due to WHO guidelines focusing on younger age groups but do not delve into a deeper analysis. 

Response: a. The authors clarify that the in-depth exploration of regional performance has been developed in subsequent works, where political, socioeconomic implications, and relationships with the development of the health system were explored. We apologize if this gap caused any difficulties in connecting the discussion, but the intention was to introduce a more robust discussion through this mapping. b. We will base our response on subsequent studies that examine oral health data from CAPP using only reliable data and compared to the literature. We have detailed the importance and implications of studies focusing on children around the age of 12. We apologize if this gap affected the study and affirm that we look forward to feedback on these responses and are fully open to considering the editor’s opinions. We appreciate the valuable contributions.

Comment 5: Study Limitations: The authors acknowledge the main limitations, such as the uneven quality of CAPP data and the use of secondary sources. However, there are additional limitations that could have been addressed more thoroughly:

• a. Lack of in-depth analysis of regional disparities: The article identifies disparities in data quality but does not propose robust solutions to address these gaps. For example, the study could have suggested methodologies to improve data collection in underrepresented regions.

• b. No validation of data from original sources: The study does not attempt to cross-validate the data included in the CAPP database with other data sources, which could have strengthened the findings. Relying solely on the CAPP database limits the study's ability to assess the true reliability of the data. 

Response: a. The authors note that methodological shortcomings are a result of underinvestment due to a lack of financial resources or prioritization, which is why new methodologies beyond detailing the most significant methodological issues were not explored. b. The authors focused exclusively on CAPP due to its significant importance. However, with the publication of evaluations from this database, we hope to develop comparative studies with other sources. The research was extremely detailed and involved evaluating various scenarios. Ideally, similar evaluations should be conducted for other data catalogs to guide researchers in choosing the database that best aligns with their study objectives and to address data quality issues, as in the case of applying the AXIS tool.

Comment 6: Implications and Practical Recommendations: Although the study provides a valuable critique of the CAPP database, it lacks actionable recommendations for policymakers. Suggestions such as implementing more rigorous inclusion criteria, improving data collection in underrepresented regions, or fostering international collaboration to enhance the database would have been beneficial.

Response: The authors attempt to address this critical need throughout the manuscript by noting that regions with significant investments have better health data. The final pages (Modified in the text, Pages 12 and 13) now include the need for stricter inclusion criteria to avoid generalizing results that impair policy development and scientific evidence, and emphasize the importance of prioritizing oral health research to foster international collaborations for better and new data.

Sincerely,

The Authors

---

## [Decision Letter · Decision Letter 1]

21 Nov 2024

Quality of evidence in the oral health international data: contributions for a global profile

PONE-D-24-29739R1

Dear Dr. Sophia Queiroz Marques dos Santos,

We’re pleased to inform you that your manuscript has been judged scientifically suitable for publication and will be formally accepted for publication once it meets all outstanding technical requirements.

Kind regards,

Hadi Ghasemi

Academic Editor

PLOS ONE

Additional Editor Comments (optional):

Reviewers' comments:

Reviewer's Responses to Questions

**Comments to the Author**

1. If the authors have adequately addressed your comments raised in a previous round of review and you feel that this manuscript is now acceptable for publication, you may indicate that here to bypass the “Comments to the Author” section, enter your conflict of interest statement in the “Confidential to Editor” section, and submit your "Accept" recommendation.

Reviewer #1: All comments have been addressed

Reviewer #3: All comments have been addressed

2. Is the manuscript technically sound, and do the data support the conclusions?

Reviewer #1: Yes

Reviewer #3: Partly

3. Has the statistical analysis been performed appropriately and rigorously? 

Reviewer #1: Yes

Reviewer #3: Yes

4. Have the authors made all data underlying the findings in their manuscript fully available?

Reviewer #1: Yes

Reviewer #3: Yes

5. Is the manuscript presented in an intelligible fashion and written in standard English?

Reviewer #1: Yes

Reviewer #3: (No Response)

6. Review Comments to the Author

Reviewer #1: Thank you for addressing my comments, I have no further feedback. Good luck with any further work in this area.

Reviewer #3: (No Response)

7. PLOS authors have the option to publish the peer review history of their article (what does this mean?). If published, this will include your full peer review and any attached files.

Reviewer #1: No

Reviewer #3: No

---

## [Editor Report · Acceptance letter]

17 Jan 2025

PONE-D-24-29739R1 

PLOS ONE

Dear Dr. Queiroz Marques dos Santos, 

I'm pleased to inform you that your manuscript has been deemed suitable for publication in PLOS ONE. Congratulations! Your manuscript is now being handed over to our production team.

Kind regards, 

on behalf of

Dr. Hadi Ghasemi 

Academic Editor

PLOS ONE